# Chemical Composition, Bioactive Compounds, and Antioxidant Activity of Two Wild Edible Mushrooms *Armillaria mellea* and *Macrolepiota procera* from Two Countries (Morocco and Portugal)

**DOI:** 10.3390/biom11040575

**Published:** 2021-04-14

**Authors:** El Hadi Erbiai, Luís Pinto da Silva, Rabah Saidi, Zouhaire Lamrani, Joaquim C. G. Esteves da Silva, Abdelfettah Maouni

**Affiliations:** 1Biology, Environment, and Sustainable Development Laboratory, Higher School of Teachers (ENS), Abdelmalek Essaâdi University, 93000 Tetouan, Morocco; Elhadi.erbiai@etu.uae.ac.ma (E.H.E.); r.saidi@uae.ac.ma (R.S.); zlamrani@uae.ac.ma (Z.L.); 2Chemistry Research Unit (CIQUP), Faculty of Sciences, University of Porto, 4169-007 Porto, Portugal; luis.silva@fc.up.pt (L.P.d.S.); jcsilva@fc.up.pt (J.C.G.E.d.S.); 3LACOMEPHI, GreenUPorto, Department of Geosciences, Environment and Territorial Planning, Faculty of Sciences, University of Porto, 4169-007 Porto, Portugal

**Keywords:** *Armillaria mellea*, *Macrolepiota procera*, bioactive compound, antioxidant activity, wild edible mushroom

## Abstract

The present study aimed to investigate the chemical composition, bioactive compounds, and antioxidant activity of two wild edible mushrooms, the honey fungus (*Armillaria mellea)* and the parasol mushroom (*Macrolepiota procera)*, collected from Northern Morocco (MA) and Portugal (PT). Those species were chosen due to their edibility, nutraceutical, and medicinal properties. Bioactive compounds (ascorbic acid, tannin, total phenolic, total flavonoid, β-carotene, and lycopene) and their antioxidant activity were determined by spectrophotometric methods. Herein, the fruiting body of the samples revealed a significantly higher amount of bioactive compounds, and values varied between the Moroccan and the Portuguese ones. Methanolic extracts shown a strong antioxidant capacity: Using DPPH free radical-scavenging activity radicals (IC_50_ 1.06–1.32 mg/mL); inhibition of β-carotene bleaching radicals (IC_50_ 0.09–0.53 mg/mL); and, reducing power radicals (IC_50_ 0.52–1.11 mg/mL). The mushroom species with the highest antioxidant capacity was *A. mellea* from MA. Chemical composition was analyzed by GC-MS and LC-MS methodologies. GC-MS analysis showed that the most abundant biomolecules group was sugar compositions in the four samples (62.90%, 48.93%, 59.00%, and 53.71%) and the main components were galactitol 16.74%, petroselinic acid 19.83%, d-galactose 38.43%, and glycerol 24.43% in *A. mellea* (MA), *A. mellea* (PT), *M. procera* (MA), and *M. procera* (PT), respectively. LC-MS analysis of individual phenolic compounds revealed that vanillic acid (198.40 ± 2.82 µg/g dry weight (dw) and cinnamic acid (155.20 ± 0.97 µg/g dw) were the main compounds detected in *A. mellea*, while protocatechuic acid (92.52 ± 0.45 and 125.50 ± 0.89 µg/g dw) was predominated in *M. procera* for MA and PT samples, respectively. In general, the results of this comparative study demonstrate that the geographic and climatic conditions of the collection site can influence biomolecule compounds and antioxidant properties of wild mushrooms. This study contributes to the elaboration of nutritional, nutraceutical, and pharmaceutical databases of the worldwide consumed mushrooms.

## 1. Introduction

Research on the wild edible mushroom of nutritional and medicinal intertest has increased greatly in recent years and is nowadays oriented towards discovering new sources beneficial to human health and with therapeutic effects on certain infectious diseases. Many edible species are economically important since they have been used both as food and medicine and contain a huge diversity of biomolecules with nutritional and biological properties such as amino, fatty and organic acids, vitamins, minerals, and sugar compositions, which have important biological roles. The mushrooms are also known for their richness of phenolic compounds, tocopherols, and carotenoids, which are considered the most responsible for antioxidant activity. In addition to antioxidant capacity, this diversity of biomolecule compounds in the wild edible mushroom is also responsible for other biological activities; namely, antibacterial, antifungal, anti-inflammatory, antitumor, and antiviral properties [1,2,3,4,5,6,7,8,9,10].

*Armillaria mellea* (Vahl: Fr.) Kummer is an edible and medicinal mushroom, commonly known as honey fungus, which belongs to the Basidiomycota phylum, Agaricales order, and Physalacriaceae family. The specimen is a saprophytic, pathogenic, and mycorrhizal fungus, and grows wildly on living and dead trees, and decaying food material. The fruiting bodies (young) of *A. mellea* is considered to be edible when thoroughly cooked. However, cases of being allergic to the honey mushroom have been reported; hence, major care should be taken when preparing and consuming it [11]. *A. mellea* has been previously reported in Morocco growing wildly under *Abies*, *Acacias*, *Amygdalus*, *Cedrus*, *Cydonia*, *Fraxinus*, *Pinus*, *Prunus,* and *Quercus* [12,13,14,15,16,17].

*Macrolepiota procera* (Scop.: Fr.) Singer is an edible basidiomycete mushroom from the order Agaricales and the family Agaricaceae, with a large, prominent fruiting body resembling a parasol. *M. procera*, commonly called the parasol mushroom, is a natural saprophytic, growing alone or in small, scattered groups in woods or at the edges of woods. This species grows wildly in forests of *Quercus*, *Cedrus,* and *Pinus* in diverse areas of Morocco including Mamora, Lalla Mimouna, Larache, and Tangier, and has been reported in Rif [13,14,16,17,18,19,20].

There are some previous studies on chemical compositions, bioactive compounds, and biological compounds of *A. mellea* and/or *M. procera* from different countries such as China [10,21], Croatia [22], India [23,24,25], Poland [26,27,28], Portugal [29,30,31,32,33], Romania [34] Serbia [35,36], Slovak Republic [37], Tanzania [38], and Turkey [39,40,41].

Nevertheless, as far as we know, in Morocco, studies on these fungi have been often qualified as descriptive, systematic, and geographical, while there is no study on the chemical compositions and biological activities. Similarly, the sector of mushroom cultivation is not very developed at the national level. In addition, this is the first report on amino acids in *A. mellea* and second in *M. procera*. In general, there are a few reports on chemical compositions of both studied mushrooms. Therefore, in current work, we were interested in studying for the first time the chemical composition, bioactive compounds, and antioxidant properties of two wild edible mushroom *A. mellea* and *M. procera* collected from biological sites in Northern Morocco to valorize the Moroccan wild mushrooms, while at the same time, making a comparative study with the ones harvested from Northern Portugal. This comparative study explored the effect of geographic locations and ecological conditions on the variability in chemical composition, the amounts of bioactive compounds, and the antioxidant capacity of studied mushrooms.

## 2. Materials and Methods

### 2.1. Standards and Reagents

N,O-Bis(trimethylsilyl)trifluoroacetamide (BSTFA), alkane standards (C_8_–C_20_ and C_21_–C_40_), meta-phosphoric acid, 2,6-dichloroindophenol sodium salt hydrate, l-ascorbic acid, (+)-catechin, vanillin reagent, Folin–Ciocalteu’s phenol reagent, (±)-6-hydroxy-2,5,7,8-tetramethylchromane-2-carboxylic acid (Trolox), β-carotene, Tween 40, linoleic acid, iron (III) chloride, sodium hydroxide, sodium nitrite, and phenolic standards, including, gallic acid, protocatechuic acid, *p*-hydroxybenzoic acid, vanillic acid, *p*-coumaric acid, ferulic acid, syringic acid, paraben, and cinnamic acid, were purchased from Sigma-Aldrich Co. (St. Louis, MO, USA). Acetonitrile, ethyl acetate, hydrochloric acid fuming 37%, pyridine, aluminum chloride, and sodium chloride were obtained from Merck KGaA (Darmstadt, Germany), and 2,2-diphenyl-l-picrylhydrazyl (DPPH) was from Alfa Aesar (Ward Hill, MA, USA). Acetone, n-hexane, and hexane were purchased from CABLO ERBA Reagent, S.A.S (Val de Reuil Cedex, France). Methanol and all other chemicals and solvents were of the highest commercial grade and obtained from Honeywell (St. Muskegon, MI, USA).

### 2.2. Mushroom Material

The fruiting bodies of *Armillaria mellea* and *Macrolepiota procera* were collected from two Mediterranean countries, one to the south in Morocco and the other to the north in Portugal. Both countries are characterized by Mediterranean-type climates. The Moroccan mushrooms, *A. mellea*, were collected during December 2017, on the *Quercus suber* tree at Talassemtane Natural Park forest in Chefchaouen region (35°10′73″ N, 05°28′35″ W—374 m of altitude, thermo-Mediterranean vegetation level, subhumid bioclimatic level at warm winter, a siliceous substrate of sandstone, acid soils), while *M. procera* were collected during January 2018, under *Acacia saligna* at Koudiat Taifour, a Biological and Ecological Interest Site (SIBE) (35°68′25″ N, 05°28′48″ W—180 m of altitude, thermo-Mediterranean vegetation level, subhumid bioclimatic level at temperate winter, a siliceous substrate of shale, acid soils) in Northern Morocco. Concerning the two Portuguese specimens, they were collected during November 2018 from Lavandeira Park in Vila Nova de Gaia in the North of Portugal (41°08′01″ N, 8°37′02″ W—50–119 m, hilltop stage of vegetation, a humid bioclimatic level at temperate at cold to very cold winter, siliceous substrate): *A. mellea* was growing on *Quercus suber* reforestation, while *M. procera* was growing on herbaceous and meadows formations.

The identification of the studied species was based on macroscopic and microscopic characterization and growing conditions, and it was made according to the two determination keys [17,42]. Mycelium of *A. mellea* and *M. procera* from Morocco were successfully cultivated on PDA (Potato Dextrose Agar) growing media at 25 °C and well preserved for further use. The fruiting bodies were immediately cleaned, weighed, cut into small pieces, air-dried, and reduced to a fine powder (20 mesh).

### 2.3. Preparation of Crude Methanolic Extract

The methanol extraction was carried out following a procedure described by [5], with slight modification. Five grams of fine dried mushroom powder (20 mesh) were extracted by stirring with 100 mL of methanol at 25 °C at 150 rpm for 24 h and filtered through Whatman N °4 paper. The residue from the filtration was extracted again twice using the procedure described earlier. The combined methanolic extracts were evaporated at 40 °C to dryness using a rotary vacuum evaporator (Rotavapor^®^ R-210, BÜCHI, Flawil, Switzerland). Then, the dried extracts were weighed and stored at −81 °C for further use. The extraction yield was calculated for each studied species.

The procedures for the spectrophotometric determination of bioactive compounds and evaluation of antioxidant activity are presented in Appendix A.

### 2.4. Phenolic Compounds Analysis by LC-MS

The analysis of phenolic compounds was carried out following a procedure previously described by the authors [43] with some modifications. Briefly, the phenolic extract was analyzed by liquid chromatography–mass spectrometry (LC-MS) (Thermo Scientific, Waltham, MA, USA). Chromatographic separation was accomplished using an Acclaim™ 120 (Thermo Scientific, Waltham, MA, USA) reverse phase C18 columns (3 µm 150 × 4.6 mm) thermostatted at 35 °C, and peaks were detected at 280 nm as the preferred wavelength. The mobile phase used was composed of 1% acetic acid in water and 100% acetonitrile. The elution gradient established was from 10% to 15% B over 5 min, from 15% to 25% B over 5 min, from 25% to 35% B over 10 min, from 35% to 50% B over 10 min, isocratic 50% B for 10 min, and re-equilibration of the column, using a flow rate of 0.5 mL/min. The identification of phenolic compounds in the samples was characterized according to their UV-Vis spectra and identified by their mass spectra and retention times in comparison with commercial standards. Quantification was made from the areas of the peaks recorded at 280 nm by comparison with calibration curves obtained from the standard of each compound. The results were expressed in µg per gram of dw.

### 2.5. GC-MS Analysis of Fruiting Body Methanolic Extract

The methanol extract of each sample (10 mg) was derivatized by adding 100 µL of anhydrous pyridine and 100 µL of N,O-Bis(trimethylsilyl)trifluoroacetamide (BSFTA), and the mixture was heated at 80 °C for 25 min, then the mixture was diluted with 200 µL of chloroform [44,45]. The derivatized solution was analyzed by using Gas Chromatography (GC) (Trace 1300 gas chromatography; Thermo Fisher Scientific, Waltham, MA, USA) linked to a mass spectrometry (MS) system (ISQ single quadrupole mass spectrometer; Thermo Fisher Scientific). The GC was equipped with a capillary column DB-5 (30 µm, 0.25 mm i.d, film thickness 0.25 µm) with a non-polar stationary phase (5% phenyl, 95% dimethylpolysiloxane). The column temperature was programmed from 50 to 350 °C at a rate of 5 °C/min. Helium was used as a carrier gas at a flow rate of 0.75 mL/min. Retention indices were calculated for all components using a homologous series of known standards of alkanes mixture (C_8_–C_20_ and C_21_–C_40_) injected in conditions equal to sample ones. Identification of components of mushroom extracts was based on retention indices (RI) relative to alkanes with those of authentic compounds and with the spectral data obtained from the databases of National Institute Standard and Technology (NIST2014) and PubChem Libraries of the corresponding compounds.

### 2.6. Statistical Analysis

Three samples were used, and all assays were carried out in triplicate. Extraction yield, bioactive compounds, and antioxidant activity values were expressed as mean ± standard deviation (SD). The statistical significance of the data was made with one-way analysis of variance (ANOVA), followed by post-hoc Tukey’s multiple comparison test with α = 0.05 using GraphPad Prism 8.0.1 software (San Diego, CA, USA).

## 3. Results and Discussion

### 3.1. Extraction Yield

As presented in Table 1, the yields of methanolic extract of the fruiting body of *A. mellea* were significantly similar for the samples from Northern Morocco (MA) (33.5%) and Portugal (PT) (32.78%) and near to the Slovak one (35.2%) [37]. Similarly for *M. procera*, there was no significant difference between the yield of MA (34.10 ± 1.7%) and PT (34.33 ± 0.96%), but they were higher than the previous studies [23,39,41].

### 3.2. Spectrophotometric Determination of Bioactive Compounds

The bioactive compounds of both samples, *A. mellea* and *M. procera*, were determined using a UV-visible spectrophotometer, and the results are presented in Table 1.

Total phenolic content (TPC) of methanolic extract of *A. mellea* from MA (20.61 mg GAE/g dme) was observed to be similar to that noted in Turkey [46] and Serbia [36], while showing a higher value in comparison with the one from PT (12.52 mg GAE/g dme). The studies from Poland [28] and Slovak Republic [37] reported a lower amount than the obtained for our studied samples. For TPC in *M. procera*, it was found that the sample from PT had significantly higher content (21.46 mg GAE/g dme) than the sample from MA (12.24 mg GAE/g dme). The value of TPC in *M. procera* from PT (in our study) was almost similar to [32] and higher than previously reported by [30].

Total flavonoid content (TFC) in *A. mellea* from PT showed significantly higher content (26.29 mg CE/g dme) in comparison with the one from MA (7.42 mg CE/g dme), while there was a small difference in *M. procera* content between both samples from MA and PT (8.61 and 10.98 mg CE/g dme). There are several studies concerning TFC in *M. procera* and the comparison values of these studies with ours are difficult because of extraction and analytical methods or ways of expressing the results.

Tannin content of both mushrooms, *A. mellea* and *M. procera*, were found in low quantities (1.37 to 1.89 mg CE/g dw), and the results showed no significant difference between samples from MA and PT.

Concerning the ascorbic acid content, the fruiting bodies provided moderated results (2.25 to 2.99 mg AAE/g dw), which were higher by about the double than the previous studies reported by the authors [31,37,39].

The amount of β-carotene and lycopene in *A. mellea* of PT was significantly higher than the MA one, while in *M. procera*, the amount of MA was higher than PT (Table 1). These results are different or near to previous studies reported by other authors [25,26,37,38,39,46].

### 3.3. Phenolic Compounds Analysis by LC-MS

The characterization of the phenolic acids and related compounds was performed by LC-MS, and the results obtained are shown in Table 2. The analysis showed a significant quantitative difference in the tested compounds between samples from MA and PT. Otherwise, the two phenolic compounds gallic acid and syringic acid in *A. mellea* from PT were not detected. Herein, vanillic acid (198.4 µg/g dw) was the main phenolic acid detected in *A. mellea* from MA followed by the related phenolic compound cinnamic acid (100.6 µg/g dw), protocatechuic acid (48.34 µg/g dw), and gallic acid (32.24 µg/g dw). Cinnamic acid was found to be the compound identified in *A. mellea* from PT with the highest amount (155.2 µg/g dw) followed by paraben (48.12 µg/g dw), protocatechuic acid (43.90 µg/g dw), and *p*-hydroxybenzoic acid (43.85 µg/g dw). By their turn, protocatechuic acid was the major component with a concentration of 93.52 and 125.5 µg/g dw, followed by cinnamic acid with a concentration of 81.93 and 90.60 µg/g dw and paraben with a concentration of 15.30 and 17.87 µg/g dw, for *M. procera* from MA and PT respectively. The compounds detected with the lowest concentration were ferulic acid (3.33 µg/g dw) and *p*-Coumaric acid (6.56 µg/g dw) in *A. mellea* from MA, *p*-coumaric acid (1.72 µg/g dw) in *A. mellea* from PT, and in *M. procera*, they were syringic acid with the values of 0.44 and 0.40 µg/g dw and *p*-Coumaric acid with the values of 2.04 and 3.07 µg/g dw from MA and PT, respectively. The concentrations of phenolic compounds in *M. procera* from MA and PT were observed to be close to each other, in particular the compounds paraben, vanillic, ferulic, and syringic acid. On the contrary, in *A. mellea*, except for protocatechuic acid, all compounds’ concentrations were highly different between MA and PT.

In the mushroom *A. mellea*, two studies from Poland [27,28] reported the detection of only one phenolic acid, which is protocatechuic acid with the values of 2.25 and 2.23 µg/g dw, respectively. However, *p*-hydroxybenzoic acid (4.00 µg/g dw) and cinnamic acid (8.67 µg/g dw) were detected in a previous study from PT by [31]. On the other hand, protocatechuic acid (5.19 µg/g dw) was the only phenolic acid detected in *M. procera* from Poland by [28], while cinnamic acid (21.53 µg/g dw) was the compound in the sample from Portugal reported previously by [30]. All these values result from previous studies were lower than our finding values.

### 3.4. GC-MS Analysis

GC-MS is a powerful tool for qualitative and quantitative analysis of various compounds present in natural products and the techniques widely used in medical, biological, and food research [47]. The GC-MS analysis of the methanolic extract of *A. mellea* MA, *A. mellea* PT, *M. procera* MA, and *M. procera* PT after its derivatization showed the presence of sixty-one, sixty-three, thirty-five, and fifty-one biomolecule compounds, respectively. These biomolecules could contribute to the medicinal quality of mushrooms (Appendix A). The identified biomolecules can be mainly divided into five main groups of compounds of each sample, namely sugars, amino acids, fatty acids, organic acids, and the fifth one being composed of the rest groups, whereas sugars were found to be the main chemical group in all samples (48.91–62.93%) (Table 3). The highest percentage constituents detected in *A. mellea* MA, *A. mellea* PT, *M. procera* MA, and *M. procera* PT were galactitol (16.74%), petroselinic acid (19.83%), d-galactose (38.43%), and glycerol (24.43%), respectively. These results demonstrate that geographic variation had a significant influence on the chemical compositions of both samples *A. mellea* and *M. procera*.

Regarding sugar compositions, as presented in Appendix A, methanolic extract of mushroom studies showed a strong quantity of sugars, whereas in *A. mellea* from MA, galactitol was a major component (16.74%), followed by threitol (10.52%), while *A. mellea* from PA was dominated by threitol (18.08%) followed by galactitol (17.21%). Concerning *M. procera,*
d-galactose (38.43%) was the main sugar detected in the sample from MA, and glycerol (24.43%) in the PT one. Previously, the main sugars observed in *A. mellea* were trehalose and mannitol reported by [31] and [36], respectively, while d-glucose, d-galactose, and d-xylose were the monosaccharides compounds detected by [21]. On the other hand, mannitol was the most abundant sugar detected in *M. procera* [29,32], while mannose was the predominate sugar in the study by [22].

Concerning amino acids, as shown in Appendix A, strong quantitative and qualitative differences in amino acids groups between MA and PT were observed in both samples. Namely, amino acids were found to be higher in *A. mellea* from MA than PT, which was contrary to *M. procera* (10.7–1.34% and 3.57–23.06%, respectively). Pidolic acid (MA; 1.39%) and alanine (PT; 0.41%) were the main compounds in *A. mellea*, while threonine (1.26%) and tryptophan (2.82%) were reported as major compounds in MA and PT, respectively. To our knowledge, there was only one study that reported the amino acid in *M. procera* by [22], which was dominated by glutamic acid.

Regarding fatty acids, there was a strong difference between *A. mellea* from PT (30%) and the other samples, *A. mellea* from MA (6.11%), *M. procera* from MA (6.37%), and PT (5%), and due to the strong quantity of the compound petroselinic acid (19.83%), they were detected in *A. mellea* from PT and were not found in the other ones. As shown in Appendix A, fatty acids of *A. mellea* from MA were dominated by linoleic acid which is in agreement with the previous investigation by [36], but contrary to [31], which found oleic acid the main fatty acid. The authors [24,29,32] reported that linoleic acid was the fattiest acid detected in *A. procera*, which is in disagreement with our samples dominated by linolelaidic acid.

For organic acids, malic acid was observed to be the main organic acid identified in *A. mellea* from MA which is in accordance with [36], while succinic acid (2.45%) was the main constituent from PA, which was almost similar to the one from MA (2.29%). Regarding *M. procera*, urea was the main compound detected *M. procera* from MA and was not detected in *M. procera* from PT, this last one being predominated by maleic acid (Appendix A). [33] found citric acid to be the main organic acid in *M. procera*.

Concerning the rest of the compounds detected by GC-MS (Appendix A), there was a difference in constituents in all of the samples and a variety of groups were observed, including steroids, alcohols, glycerides, nucleosides, etc. Ergosterol and adenosine were the two main compounds detected in methanolic extracts of *A. mellea* and *M. procera* from both countries MA and PT.

### 3.5. Antioxidant Activity

In the current investigation, the antioxidant activities of methanolic extracts of *A. mellea* and *M. procera* collected from MA and PT were determined spectrophotometrically using three different assays: DPPH radical scavenging, β-carotene/linoleate, and ferricyanide/Prussian blue activity. The results of antioxidant activities expressed in IC_50_ values are presented in Table 4. These results showed a very significant difference between the antioxidant capacity of the methanolic extracts of all samples and Trolox, a standard that was used as a control.

Regarding DPPH radical-scavenging activity (Appendix A), the in vitro model of scavenging stable DPPH free radicals can be used to evaluate the antioxidative activities in a relatively short time. The methanol extracts of the samples were able to reduce the stable free radical DPPH to the yellow-colored DPPH, which was due to the high content of total phenols and flavonoids of the samples. The best free radical scavenging activity was extracted by *A. mellea* from MA (1.06 mg/mL), which showed a low significant difference with other sample extracts. However, these results were similar to a previous study by [32], and were higher than reported investigation by [28,29,37].

Regarding β-carotene-linoleate bleaching assay (Appendix A)***,*** the methanol extracts of the samples were able to inhibit the discoloration of β-carotene and have shown an important antioxidant capacity. There was a significant difference between all samples, whereas *A. mellea* extract from MA showed a higher antioxidant effect than the one from PT. On other hand, *M. procera* from PT showed the highest inhibition of linoleic acid oxidation (0.09 mg/mL) at 120 min of incubation. Previously, [29,32] reported a low antioxidant capacity of methanolic extract of *M. procera* than that of our samples. These considerable results may be correlated with the high content of carotenoids in the methanolic extracts of the investigated mushrooms.

Regarding reducing power by ferricyanide/Prussian blue assay (Appendix A), the reducing power method reflects the electron donation ability of antioxidants present in the extracts to convert Fe^3+^ into Fe^2+^. The amount of the Fe^2+^ complex was followed by measuring the formation of Perls’ Prussian blue at the absorbance of 690 nm. As shown in Table 4 and Appendix A, the results of reducing power assay were significantly different in the comparison between samples from MA and PT, where *A. mellea* from MA showed a stronger reducing power (IC_50_ = 0.52 mg/mL), while the lowest one was observed in *M. procera* from PT (IC_50_ = 1.11 mg/mL). The results from our *M. procera* (MA and PT) were higher than previously reported studies by [29,32]. This important result of reducing power could be due to the ability of different chemical compositions and one of its major compounds to reduce Fe^3+^.

Overall, our findings showed the investigated mushrooms did not have a strong significant difference in their antioxidant activities and IC_50_ values for each sample were not highly influenced by the growing conditions (*p* < 0.05).

## 4. Conclusions

In this study, for the first time, the two wild edible *A. mellea* and *M. procera* from Morocco were investigated for their chemical composition, bioactive compounds, and antioxidant activities and compared with the ones from Portugal, which is a location characterized by different geographic, micro-ecological conditions and with a strong Atlantic influence, compared to Morocco with a strong Mediterranean influence. The fruiting bodies of both samples from MA and PT showed an important amount of bioactive compounds and a significant difference between each other. A total of more than 60 and 35 compounds have been identified in derivatized methanol extracts of *A. mellea* and *M. procera*, respectively, using GC-MS analysis.

Results of antioxidant activities by the three assays DPPH radical scavenging activity, inhibition of β-carotene bleaching, and ferric reducing power showed a strong antioxidant capacity that can be due to the samples, which are rich in bioactive and chemical compounds. Statistically, the antioxidant results were observed to be a small difference between mushrooms from MA and PT.

In general, the observed experimental data in this comparative study demonstrate clearly that the chemical composition, bioactive compounds, and antioxidant properties of the wild mushrooms can be affected by the geographic, microclimatic, and edaphic conditions of the collection site. However, this investigation contributes to the elaboration of nutritional and pharmaceutical databases of the worldwide consumed mushrooms.

## Figures and Tables

**Table 1 biomolecules-11-00575-t001:** Extraction yield and bioactive compound contents in the dried fruiting body of mushroom studies ^1^.

Bioactive Compounds		*A. mellea*	*M. procera*
Extraction yield (%)	MA	33.5 ± 1.25	34.10 ± 1.7
	PT	32.78 ± 1.67	34.33 ± 0.96
Total phenolic (mg GAE/g dme)	MA	20.61 ± 0.18 ^b^	12.24 ± 0.27 ^b^
	PT	12.52 ± 0.55 ^c^	21.46 ± 0.22 ^a^
Total flavonoid (mg CE/g dme)	MA	7.42 ± 0.21 ^d^	8.61 ± 0.13 ^d^
	PT	26.29 ± 0.52 ^a^	10.98 ± 0.13 ^c^
Tannin (mg CE/g dw)	MA	1.49 ± 0.03 ^f^	1.31 ± 0.03 ^g^
	PT	1.37 ± 0.07 ^f^	1.89 ± 0.07 ^f^
Ascorbic acid (mg AAE/g dw)	MA	2.55 ± 0.05 ^e^	2.25 ± 0.14 ^e^
	PT	2.99 ± 0.04 ^e^	2.45 ± 0.11 ^e^
β-carotene (µg/g dme)	MA	0.08 ± 0.002 ^c^*	0.44 ± 0.003 ^a^*
	PT	0.58 ± 0.003 ^a^*	0.32 ± 0.001 ^c^*
Lycopene (µg/g dme)	MA	0.05 ± 0.003 ^d^*	0.29 ± 0.004 ^b^*
	PT	0.39 ± 0.004 ^b^*	0.20 ± 0.002 ^f^*

^1^ Values are expressed as means ± SD of three independent measurements. According to Duncan’s multiple range test, means within a column followed by the same lowercase letter are not significantly different at *p* < 0.05 (homogeneous groups). The statistical study of β-carotene and lycopene was done separately *.

**Table 2 biomolecules-11-00575-t002:** Phenolic acids and related compound characterized by LC-MS ^1^.

N°.	Phenolic Compounds	*A. mellea* (µg/g dw)		*M. procera* (µg/g dw)	
		MA	PT	MA	PT
1	Cinnamic acid	100.60 ± 0.21 ^b^	155.20 ± 0.97 ^a^	81.93 ± 0.36 ^b^	90.60 ± 0.50 ^b^
2	Ferulic acid	3.33 ± 0.04 ^h^	18.52 ± 0.13 ^e^	5.11 ± 0.25 ^e^	6.36 ± 0.16 ^e^
3	Gallic Acid	32.24 ± 0.45 ^d^	0.00 ^f^	11.58 ± 0.49 ^d^	16.41 ± 0.03 ^c^
4	Paraben	17.40 ± 1.90 ^e^	48.12 ± 1.75 ^b^	15.30 ± 0.66 ^c^	17.87 ± 0.37 ^c^
5	*p*-Coumaric acid	6.56 ± 0.37 ^g^	1.72 ± 0.29 ^f^	2.04 ± 0.04 ^ef^	3.07 ± 0.16 ^f^
6	*p*-Hydroxybenzoic acid	13.00 ± 0.04 ^f^	43.85 ± 0.51 ^f^	9.99 ± 0.66 ^d^	16.76 ± 0.28 ^c^
7	Protocatechuic acid	48.34 ± 0.33 ^c^	43.90 ± 0.24 ^c^	93.52 ± 0.45 ^a^	125.5 ± 0.89 ^a^
8	Syringic acid	7.80 ± 0.33 ^g^	0.00 ^f^	0.44 ± 0.019 ^f^	0.40 ± 0.06 ^g^
9	Vanillic acid	198.4 ± 2.82 ^a^	38.02 ± 0.61 ^d^	8.61 ± 0.29 ^d^	8.42 ± 0.39 ^d^

^1^ Each value is expressed as means ± SD (n = 3). According to Duncan’s multiple range test, means within a column followed by the same lowercase letter are not significantly different at *p* < 0.05 (homogeneous groups).

**Table 3 biomolecules-11-00575-t003:** Biomolecules groups of methanolic extracts analysis by GC-MS.

Compound Names	*A. mellea* (%)		*M. procera* (%)	
	MA	PT	MA	PT
Sugar compositions	62.90	48.93	59.00	53.71
Amino acids	10.7	1.34	3.57	23.06
Fatty acids	6.11	30	6.37	5
Organic acids	11.2	7.78	21	7.44
Other groups	9.07	11.94	10.06	10.78
Total	99.98	99.99	100	99.99

**Table 4 biomolecules-11-00575-t004:** IC_50_ (mg/mL) of antioxidant properties of the methanolic extracts from Northern Morocco (MA) and Portugal (PT).

Assays			*A. mellea*	*M. procera*
A.	DPPH radical-scavenging activity	MA	1.06 ± 0.12 *	1.19 ± 0.05 ^ns^
		PT	1.32 ± 0.09 *	1.31 ± 0.05 ^ns^
B.	β-carotene/linoleate assay	MA	0.43 ± 0.01 ***	0.16 ± 0.03 **
		PT	0.53 ± 0.01 ***	0.09 ± 0.02 **
C.	Ferricyanide/Prussian blue assay	MA	0.52 ± 0.02 ***	0.96 ± 0.01 ***
		PT	0.62 ± 0.01 ***	1.11 ± 0.02 ***

The results are presented as mean ± SD (n = 3). ^ns^
*p* > 0.05; * *p* ≤ 0.05, ** *p* ≤ 0.01, and *** *p* ≤ 0.001 indicate significant differences between samples from MA and PT.

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
