# Peer review of "Chemical Composition, Bioactive Compounds, and Antioxidant Activity of Two Wild Edible Mushrooms Armillaria mellea and Macrolepiota procera from Two Countries (Morocco and Portugal)"

_biomolecules, 2021, doi:10.3390/biom11040575_

Round 1

Reviewer 1 Report

In the manuscript “Chemical composition, bioactive compounds and antioxidant activity of two wild edible mushrooms Armillaria mellea and Macrolepiota procera from two countries (Morocco and Portugal)”, the results themself are interesting, some comments on the article are:

  1. The information on the methodology that is online 128-130 (on the isolation of two species) is relevant for the article.
  2. Check appointment format (line 76)
  3. The appointment number 43 is in line 151 and later it is referred to the appointment 46 (line 217), but, I did not find the appointment 44 and 45.
  4. Check the tables, some numbers have commas and others have a point.
  5. The names of the countries were abbreviated in the abstract Morocco (MA) and Portugal (PT), but in the following sections in some cases they put the full name, in addition, in table 1 and 4 and in results, they return to abbreviate.
  6. Review the format of the journal and homogenize, in some tables the information of the statistical analysis is this after the table (table 1, 2) and in table 4 they put said information after the title.
  7. The authors indicate that the work was: a comparative study that explores the effect of geographical location and ecological conditions on the variability in the chemical composition, the amounts of bioactive compounds and the antioxidant capacity of the fungi studied. Differences in the composition of fruiting bodies of the same species, due to biotic and abiotic factors, have been widely reported. However, at work, the effect is not clear, I suggest that they could give more information about the collection sites: vegetation, climate, geology, soil (pH, organic matter composition, minerals), etc. and relate it to the differences found between the species from both collection sites.
  8. The work makes a contribution to the knowledge of two species of wild mushrooms, which is important in the elaboration of nutritional, nutraceutical and pharmaceutical databases of the mushrooms consumed worldwide (as indicated in the conclusion), therefore They must assess whether they are really relating their results to geographic and ecological factors.

Author Response

1. The information on the methodology that is online 128-130 (on the isolation of two species) is relevant for the article.

Okay. Thanks

2. Check appointment format (line 76)

Okay.

This species growing wildly in forests of Quercus, Cedrus, and Pinus in diverse areas of Morocco including Mamora, Lalla Mimouna, Larache, Tangier and reported in Rif [13,14,16–20]

3. The appointment number 43 is in line 151 and later it is referred to the appointment 46 (line 217), but, I did not find the appointment 44 and 45.

  • 43 in line 151 is methodology : The analysis of phenolic compounds was carried out following a procedure previously described by the authors [43: Reis, F.S.; Martins, A.; Barros, L.; Ferreira, I.C.F.R. Antioxidant Properties and Phenolic Profile of the Most Widely Appreciated Cultivated Mushrooms: A Comparative Study between in Vivo and in Vitro Samples. Food Chem. Toxicol. 2012, 50, 1201–1207, doi:10.1016/j.fct.2012.02.013] with some modifications.
  • 46 in line 217 is result: Total Phenolic Content (TPC) of methanolic extract of mellea from MA (20.61 mg GAE/g dme) was observed to be similar to that noted in Turkey [46 Aytar, E.C.; Akata, I.; Açik, L. Antioxidant and Antimicrobial Activities of Armillaria Mellea and Macrolepiota Procera Extracts. 2020, 8].
  • The appointment numbers 44 and 45 are in line 173:

The methanol extract of each sample (10 mg) was derivatized by adding 100 µL of anhydrous pyridine and 100 µL BSTFA (N,O-Bis(trimethylsilyl)trifluoroacetamide), and the mixture was heated at 80°C for 25 min, then the mixture was diluted with 200 µL chloroform [44, 45].

4. Check the tables, some numbers have commas and others have a point.

Okay, It is done.

5. The names of the countries were abbreviated in the abstract Morocco (MA) and Portugal (PT), but in the following sections in some cases, they put the full name, in addition, in table 1 and 4 and in results, they return to abbreviate.

Okay. It is done

As presented in Table 1, the yields of methanolic extract of the fruiting body of A. mellea was significantly similar for the samples from MA (33.5%) and PT (32.78%) and near to the Slovak one (35.2%) [37].

Table 1. Extraction yield and bioactive compound contents in the dried fruiting body of mushroom studies (MA – Morocco; PT - Portugal). 1

In the current investigation, the antioxidant activities of methanolic extracts of A. mellea and M. procera collected from Morocco (MA) and Portugal (PT)

Table 4. IC50 (mg/ml) of Antioxidant properties of the methanolic extracts from Morocco (MA) and Portugal (PT).

6. Review the format of the journal and homogenize, in some tables the information of the statistical analysis is this after the table (table 1, 2) and in table 4 they put said information after the title.

Okay. It is done

7. The authors indicate that the work was: a comparative study that explores the effect of geographical location and ecological conditions on the variability in the chemical composition, the amounts of bioactive compounds and the antioxidant capacity of the fungi studied. Differences in the composition of fruiting bodies of the same species, due to biotic and abiotic factors, have been widely reported. However, at work, the effect is not clear, I suggest that they could give more information about the collection sites: vegetation, climate, geology, soil (pH, organic matter composition, minerals), etc. and relate it to the differences found between the species from both collection sites.

The fruiting bodies of Armillaria mellea and Macrolepiota procera were collected from two Mediterranean countries, one to the south in Morocco and the other to the north in Portugal. Both countries are characterised by Mediterranean-type climates. The Moroccan mushrooms, A. mellea were collected during December 2017, on the Quercus suber tree at Talassemtane Natural Park forest in Chefchaouen region (35°10’73’’ N, 05°28’35’’ W - 374 m of altitude, thermo-Mediterranean vegetation level, Subhumid bioclimatic level at warm winter, a siliceous substrate of sandstone, acid soils), while, M. procera were collected during January 2018, under Acacia saligna at Koudiat Taifour, a Biological and Ecological Interest Site (SIBE) (35°68’25’’ N, 05°28’48’’ W - 180 m of altitude, thermo-Mediterranean vegetation level, Subhumid bioclimatic level at temperate winter, a siliceous substrate of shale,  acid soils ) in Northern Morocco. Concerning the two Portuguese specimens, they were collected during November 2018, from Lavandeira Park in Vila Nova de Gaia in the North of Portugal (41°08’01” N, 8°37’02” W – 50-119 m, hilltop stage of vegetation, a humid bioclimatic level at temperate at cold à very cold winter, siliceous substrate): A. mellea was growing on Quercus suber reforestation, while, M. procera were growing on herbaceous and meadows formations.

8. The work makes a contribution to the knowledge of two species of wild mushrooms, which is important in the elaboration of nutritional, nutraceutical and pharmaceutical databases of the mushrooms consumed worldwide (as indicated in the conclusion), therefore They must assess whether they are really relating their results to geographic and ecological factors.

Conclusion

In this study, for the first time, the two wild edible A. mellea and M. procera from Morocco have investigated their chemical composition, bioactive compounds, and antioxidant activities and compared with the ones from Portugal, which is a location characterized by different geographic, micro ecological conditions and with a strong Atlantic influence than Morocco with a strong Mediterranean influence. In general, the observed experimental data in this comparative study demonstrate clearly that the chemical composition, bioactive compounds, and antioxidant properties of the wild mushrooms can be affected by the geographic, microclimatic, and edaphic conditions of the collection site. However, this investigation contributes to the elaboration of nutritional, nutraceutical, and pharmaceutical databases of the worldwide consumed mushrooms.

Reviewer 2 Report

This ms describes the bioactive compounds and antioxidant activity of two wild edible mushrooms, Armillaria mellea and Macrolepiota procera. The data are enumerative and the new finding of this study is unclear. Are there any characteristic compound(s) found in these mushrooms? Technically methanol is not an appropriate solvent for the quantitative analysis of beta-carotene.

Author Response

  • This ms describes the bioactive compounds and antioxidant activity of two wild edible mushrooms, Armillaria mellea and Macrolepiota procera. The data are enumerative and the new finding of this study is unclear. Are there any characteristic compound(s) found in these mushrooms? Technically methanol is not an appropriate solvent for the quantitative analysis of beta-carotene.

I agree if our research had as the main goal the research of new compounds, but for us, it is a comparative study of chemical compounds and antioxidant activity between mushroom species taken from Morocco and Portugal. And it appeared that there are differences in chemical composition and activity for the same species on both sides of the Mediterranean. Also, to our knowledge, besides a paper from us before this, there is no study on the chemical compositions and biological activities of Moroccan wild mushrooms, which make us more interesting to valorize Moroccan mushrooms and discover if they are any new important compounds.  

  • Technically methanol is not an appropriate solvent for the quantitative analysis of beta-carotene.

Okay. I agree with you, but Methanol was used only for extraction but not in the assay. We also adopted the methods described by Barros et al., 2007.

« β-Carotene and Lycopene Content were determined following a method previously described by the authors [5,6]. 100 mg of dried methanol extract was vigorously shaken with 10 ml of acetone-hexane (4:6) for 1 min and filtered through Whatman N° 4 filter paper. The filtrate solution absorbance (A) was measured at 453, 505, 645, and 663 nm using a UV-Vis spectrophotometer.

Reviewer 3 Report

Lines 45-56 should be better discussed.

The aim and the novelty character of paper should be better described

A graphical scheme should be inserted for 2.3. Preparation of Crude Methanolic Extract

The methodological procedure for phenols and antioxidant should be moved for Supplementary section to material and methods of main text and introcuctive lines on study of antioxidant approach should be added such as:

Durazzo A. Study Approach of Antioxidant Properties in Foods: Update and Considerations. 02/2017; 6(3):17., DOI:10.3390/foods6030017

Table 2 should be better described in the text.

Author Response

  • Lines 45-56 should be better discussed.

Okay. It is done

Research on the wild edible mushroom of nutritional, nutraceutical, and medicinal intertest has increased greatly in recent years and is nowadays oriented towards discovering new sources beneficial to human health and with therapeutic effects on certain infectious diseases. Many edible species are economically important since have been used both as food and medicine and they contain a huge diversity of biomolecules with nutritional and biological properties such as amino, fatty and organic acids, vitamins, minerals, and sugar compositions which has important biological roles. The mushrooms are also, known for their richness of phenolic compounds, tocopherols, and carotenoids which are considered to be the most responsible for antioxidant activity. Besides antioxidant capacity, this diversity of biomolecule compounds in the wild edible mushroom are responsible also for other biological activities, namely, antibacterial, antifungal, anti-inflammatory, antitumor, and antiviral properties [1–10].   they are rich in proteins, carbohydrates, vitamins, minerals, and low in fats, also they are a good source of amino acids, fatty acids, organic acids, sugar compositions and contain nutraceutical compounds such as phenolic compounds, tocopherols, ascorbic acid, and carotenoids. These nutrients and biomolecule compounds are responsible for the biological activity (antioxidant, antimicrobial, anti-inflammatory, antitumor, and antiviral effects)

  • The aim and the novelty character of paper should be better described

Okay. It is done

Nevertheless, as far as we know, in Morocco, studies on these fungi have been often qualified as descriptive, systematic, and geographical, while there is no study on the chemical compositions and biological activities. Similarly, the sector of mushroom cultivation is not very developed at the national level. Also, this is the first report on amino acids in A. mellea and second in M. procera, in general, there are a few reports on chemical compositions of both studied mushrooms. Therefore, in current work, we are interested in studying for the first time the chemical composition, bioactive compounds, and antioxidant properties of two wild edible mushroom A. mellea and M. procera collected from biological sites in northern Morocco to valorize the Moroccan wild mushrooms, in the same time, to make a comparative study with the ones harvested from northern Portugal. This comparative study will explore the effect of geographic locations and ecological conditions on the variability in chemical composition, the amounts of bioactive compounds, and the antioxidant capacity of studied mushrooms.

  • A graphical scheme should be inserted for 2.3. Preparation of Crude Methanolic Extract

Okay: We can propose this graphic scheme, but it does not seem to us to be justified enough in the article. However, it is possible to add it in the supplement if you see its importance.

The methodological procedure for phenols and antioxidant should be moved for Supplementary section to material and methods of main text and introcuctive lines on study of antioxidant approach should be added such as:

Durazzo A. Study Approach of Antioxidant Properties in Foods: Update and Considerations. 02/2017; 6(3):17., DOI:10.3390/foods6030017

Okay : Spectrophotometric Determination of Bioactive Compounds (Total Phenolic Content) and Evaluation of Antioxidant Activity are alredy in Supplementary section. It seems appropriate to leave the Phenolic Compounds Analysis by LC-MS part in the article as it is, given its importance.

  • Table 2 should be better described in the text.

Okay. It is done

The characterization of the phenolic acids and related compounds was performed by LC-MS, and the results obtained are shown in Table 2. The analysis showed a significant quantitative difference in the tested compounds between samples from MA and PT. Otherwise, it were not detected the two phenolic compounds gallic acid and syringic acid in A. mellea from PT. Herein, vanillic acid (198.4 µg/g dw) was the main phenolic acid detected in A. mellea from MA followed by the related phenolic compound cinnamic acid (100.6 µg/g dw), protocatechuic acid (48.34 µg/g dw), and gallic acid (32.24 µg/g dw). Cinnamic acid was found to be the compound identified in A. mellea from PT with the highest amount (155.2 µg/g dw) followed by paraben (48.12 µg/g dw), protocatechuic acid (43.90 µg/g dw), and p-hydroxybenzoic acid (43.85 µg/g dw). By their turn, protocatechuic acid was the major component with a concentration of 93.52 and 125.5 µg/g dw, followed by cinnamic acid with a concentration of 81.93 and 90.60 µg/g dw and paraben with a concentration of 15.30 and 17.87 µg/g dw, for M. procera from MA and PT respectively. The compounds detected with the lowest concentration were ferulic acid (3.33 µg/g dw) and p-Coumaric acid (6.56 µg/g dw) in A. mellea from MA, p-coumaric acid (1.72 µg/g dw) in A. mellea from PT and syringic acid in M. procera with the value of 0.44 µg/g dw from MA and 0.40 µg/g dw from PT.  and in M. procera, were syringic acid with the values of 0.44 and 0.40 µg/g dw and p-Coumaric acid with the values of 2.04 and 3.07 µg/g dw from MA and PT, respectively. The concentrations of phenolic compounds in M. procera from MA and PT were observed to be closed to each other, in particular the compounds paraben, vanillic, ferulic, and syringic acid, in contrary, A. mellea except protocatechuic acid all compounds’ concentrations were highly different between MA and PT.

Round 2

Reviewer 2 Report

beta-Carotene is very sparingly sol in methanol.   https://pubchem.ncbi.nlm.nih.gov/compound/beta-Carotene#section=Solubility   The quantitative analysis of beta-carotene must be done using an  appropriate solvent for the extraction.    

Author Response

Okay Dear, we fully agree with you and In future works, we will consider your suggestions and we will be using another appropriate solvent for beta-carotene determination. However, we still try to explain our bad choice of Methanol and we wish you to accept: The quantitative analysis of beta-carotene in our present work was according to [1] which they used methanol as a solvent of extraction, and we found many works that used the same methodology described by [1] and they found a considerable amount of beta carotene in their samples, we can mention some of them here [2–7], for that and because of methanol is more available and less toxic we chose this methodology of beta-carotene determination, however, according to your cited reference “PubChem”, beta-Carotene is very sparingly sol in methanol, the same, they are some works concluded that the content of beta-carotene content in methanol is lower than some of the other solvents, also according to the literature [8], the solubility of beta-carotene in methanol is 10 mg/L.

Reference:

  1. Barros, L.; Ferreira, M.-J.; Queirós, B.; Ferreira, I.C.F.R.; Baptista, P. Total Phenols, Ascorbic Acid, β-Carotene and Lycopene in Portuguese Wild Edible Mushrooms and Their Antioxidant Activities. Food Chem. 2007, 103, 413–419, doi:10.1016/j.foodchem.2006.07.038.
  2. Pavithra, M.; Sridhar, K.R.; Greeshma, A.A.; Tomita-Yokotani, K. Bioactive Potential of the Wild Mushroom Astraeus Hygrometricus in South-West India. Mycology 2016, 7, 191–202, doi:10.1080/21501203.2016.1260663.
  3. Jaworska, G.; Pogoń, K.; Bernaś, E.; Duda‐Chodak, A. Nutraceuticals and Antioxidant Activity of Prepared for Consumption Commercial Mushrooms Agaricus Bisporus and Pleurotus Ostreatus. J. Food Qual. 2015, 38, 111–122, doi:https://doi.org/10.1111/jfq.12132.
  4. Tel, G.; Ozturk, M.; Duru, M.E.; Turkoglu, A. Antioxidant and Anticholinesterase Activities of Five Wild Mushroom Species with Total Bioactive Contents. Pharm. Biol. 2015, 53, 824–830, doi:10.3109/13880209.2014.943245.
  5. Kuka, M. BIOACTIVE COMPOUNDS IN LATVIAN WILD EDIBLE MUSHROOM BOLETUS EDULIS. 2011, 5.
  6. Robaszkiewicz, A.; Bartosz, G.; Ławrynowicz, M.; Soszyński, M. The Role of Polyphenols, β-Carotene, and Lycopene in the Antioxidative Action of the Extracts of Dried, Edible Mushrooms. J. Nutr. Metab. 2010, 2010, doi:10.1155/2010/173274.
  7. Barros, L.; Cruz, T.; Baptista, P.; Estevinho, L.M.; Ferreira, I.C.F.R. Wild and Commercial Mushrooms as Source of Nutrients and Nutraceuticals. Food Chem. Toxicol. 2008, 46, 2742–2747, doi:10.1016/j.fct.2008.04.030.
  8. Craft, N.E.; Soares, J.H. Relative Solubility, Stability, and Absorptivity of Lutein and .Beta.-Carotene in Organic Solvents. J. Agric. Food Chem. 1992, 40, 431–434, doi:10.1021/jf00015a013.